# ON UNDERSTANDING DENOISING CAPABILITY IN HYPERGRAPH REPRESENTATION LEARNING

## ABSTRACT

Hypergraphs are a powerful model for high-order relations and group interactions among entities. While many real-world network instances modeled by hypergraphs, e.g., social networks, brain connectome networks, and online question-answering communities, are rich in noise and error-prone, existing hypergraph representation learning methods often assume that hypergraphs contain limited or no noise. We reveal that even a small amount of Gaussian noise can deteriorate the performance in node classification and hyperedge prediction. In this paper, we study the problem of alleviating the impact of noises present in node features on hypergraph representation learning. We first establish the connection between receptive fields and denoising capabilities, showing increasing receptive fields may enhance the denoising ability and robustness. We then develop a four-stage message-passing method that can increase the receptive fields within a single neural network layer, which is applicable to any existing two-stage SOTA methods. We demonstrate the increase in receptive fields both theoretically and empirically. We have performed extensive experiments, including analysis of convergence time, an ablation study, and visualization of node embeddings to verify that our four-stage enhanced models achieve superior performance in node classification and hyperedge prediction under various noise settings.

## 1 INTRODUCTION

Hypergraphs are a generalization of graphs by allowing a hyperedge to connect more than two nodes, providing strong expressive ability for higher-order relations among real-world entities. Hypergraph representation learning aims to find vectorized latent representations for nodes, hyperedges, or hypergraphs, which can then be used for solving downstream application-specific analytical tasks. Hypergraph representation learning has found wide practical applications Benson in social networks, collaboration networks, online question-answering communities, brain connectome networks, biological networks Sanchez-Gorostiaga et al. (2019), and ecology Grilli et al. (2017). In the classic clique expansion method Feng et al. (2019), pairwise nodes in a hyperedge are simply connected to transform a hypergraph into a graph. However, this method may lose critical higher-order information inherent in the original hypergraph, resulting in suboptimal representation and downstream learning and analytical tasks. Effectively, accurately, directly capturing the essence of these high-order relationships is crucial for learning and analyzing complex hypergraphs.

Recently, there has been a growing interest in using Hypergraph Neural Network (HGNN) approaches for hypergraph representation learning. Existing HGNN models can be classified into two types: single-stage and two-stage hypergraph message passing. The former converts a hypergraph into a graph, such as through clique expansion Agarwal et al. (2005) or line expansion Yang et al. (2022), and then applies well-established Graph Neural Networks (GNNs), such as GCN Kipf & Welling (2016) or GIN Xu et al. (2018), in the transformed graph for representation learning. This category includes early methods such as HGNN Feng et al. (2019), HyperGCN Yadati et al. (2019), and line expansion Yang et al. (2022). Their effectiveness highly depends on how the transformed graph preserves the high-order information from the original hypergraph, and they often do not explicitly learn the hyperedge representation.

In contrast, two-stage hypergraph message-passing models explicitly obtain the representation of hyperedges for more comprehensive node/hypergraph representations. The two stages can be

essentially described by first passing messages from nodes to hyperedges containing the nodes, and then hyperedges passing messages back to nodes in the hyperedges. UniGNN Huang & Yang (2021) proposed a unified framework for interpreting the message-passing process in graph and hypergraph neural networks, and generalized several GNNs for hypergraphs. AllSetTransformer Chien et al. (2022) considered hypergraph neural networks as compositions of multiset functions and connected with recent advances in deep learning of multiset functions. Recently, ED-HNN Wang et al. (2023) introduced equivariant hypergraph diffusion algorithms to the two-stage method, achieving good results while being tolerant to the oversmoothing.

All these HGNNs made an implicit assumption that input hypergraphs contain limited or zero noise and have a high signal-to-noise ratio. However, many real-world hypergraph network data, such as social networks, online question-answering communities, brain connectome networks, and biological networks, are rich in noise and error-prone. Commonly-seen noises include node feature noise, missing groundtruth hyperedges, or spurious hyperedges. The latter two can be seen in the construction of brain connectivity networks based on Pearson's correlation in medical imaging data Li et al. (2019): true brain interactions among multiple brain regions were not captured (false negative) or spurious interactions were added (false positive). In contrast, possibly the most common type of noises is noises in node features, which are typically random samples drawn from a Gaussian distribution. They are also known as classical measurement error Carroll et al. (2006), where the truth is measured with an additive error. Such noises are ubiquitous, can have different means and variances depending on application scenarios, and may result in suboptimal performance by hindering the capture of true high-order relations. Unfortunately, existing HGNN literature has overlooked the existence of such noises in node features and its robustness against the noises. In our experiments where we added Gaussian noise to node features in widely-used benchmarking hypergraphs Cora and DBLP-CA Yadati et al. (2019), state-of-the-art HGNN methods UniGNN, AllSetTransformer and ED-HNN all suffered significant accuracy degradation of about 10% to 39% in node classification, rendering them considerably less reliable and desirable.

In this paper, we study the problem of alleviating the impact of noises present in node features on hypergraph representation learning through a four-stage hypergraph neural network framework. We establish the connection between receptive fields and denoising capability and show that increasing receptive fields may enhance the denoising ability and robustness. We then develop a method for effectively increasing receptive fields within a single neural network layer. Our main contributions are summarized as follows.

- We carry out the first study on how to alleviate the impact of noises present in node features on hypergraph representation learning. We highlight that even a small amount of Gaussian noise can result in significant accuracy degradation in node classification and hyperedge prediction.

- We establish the connection between receptive fields of a hypergraph message-passing model and its denoising capability, and conjecture that increasing the receptive fields can enhance the denoising capability.

- We develop a four-stage hypergraph message-passing framework to enhance the receptive fields within a single neural network layer. This approach is applicable to all existing two-stage message-passing methods, as it adds two new stages to them: hyperedge communication and node communication. We demonstrate the increase in receptive fields both theoretically and empirically.

- Finally, our extensive experiments confirm that our models achieve superior performance in node classification and hyperedge prediction under various noise settings, including non-zero mean Gaussian, ablation, and visualization of node embeddings. Despite the added complexity, the convergence times of our models are similar to their two-stage counterparts.

## 2 RELATED WORK

**Hypergraph Learning** To learn accurate hypergraph embeddings, HGNN Feng et al. (2019) first introduced clique expansion, inspiring several follow-up works Dong et al. (2020); Bai et al. (2021); Xia et al. (2021). Instead of connecting all nodes in a hyperedge, HyperGCN Yadati et al. (2019) only connects a linear number of selected nodes via the hypergraph Laplacian matrix Chan et al. (2018).

Later, HyperSAGEArya et al. (2020) employed a two-stage message-passing neural network through star expansion, treating each hyperedge as a node, and connecting nodes within that hyperedge to this node. Based on the two-stage procedure, UniGNNHuang & Yang (2021) developed a unified framework for GNNs and HGNNs. It also proposed a variant of the 1-dimensional Generalized Weisfeiler-Leman Algorithm (1-GWL) for hypergraph isomorphism testing following Böker (2019) and bounded the expressive power of UniGNNs for hypergraphs. AllSetTransformer Chien et al. (2022) unified a class of two-stage models with multiset functions, employing them to learn the most adequate message-passing mechanism for each dataset and task. Inspired by hypergraph diffusion algorithms, Wang et al. (2023) proposed ED-HNN that can approximate arbitrary continuous permutation-equivariant diffusion operators. Specifically, based on two-stage methods it computes specialized hyperedge-to-node messages for each node in a hyperedge. Feng et al. (2024) proposed hypergraph learning using hypergraph kernel tricks, such as subtree and hyperedge kernels, adapted from traditional graph kernel methods. A more comprehensive discussion of hypergraph representation learning can be found in the survey Antelmi et al. (2023). However, these methods are sensitive to node feature noise, and robust learning under noise remains an open problem.

**Feature Denoising** The impact of noisy node features on graph learning has been studied primarily in the context of GNNs Wu et al. (2019); Liu et al. (2019); Zügner et al. (2020); Li et al. (2022a). , to the best of our knowledge. Nettack Zügner et al. (2020) showed that even minor feature noise can greatly degrade performance on graph-structured data. PEEGA Li et al. (2022a) proposed a method to measure the negative impact of various adversarial attacks based on node representations, formulating a single-level optimization for efficient computation. There is also a body of work focusing on adversarial defense techniques that improve robustness against such attacks. Many methods have been proposed, including preprocessing the input graph Entezari et al. (2020); Li et al. (2022a); Wu et al. (2022); Li et al. (2022b); Song et al. (2022), adding trainable edge weights Wu et al. (2019), test-time graph transformation Jin et al. (2022), node location anonymization Liu et al. (2022), and utilizing attention mechanisms Zhu et al. (2019), have been present to improve the robustness against adversarial attacks. Zügner & Günnemann (2019) provided certifiable (non-)robustness for graph nodes with respect to perturbations of the node attributes, meaning that a certified node is guaranteed to be robust under any possible perturbation. In contrast, the non-robust case indicates no such guarantee. Gosch et al. (2023) studied changes to graph structures aware of semantic content changes (i.e., changed ground-truth labels), relaxing the assumption of unchanged semantic content. They introduced the concept of over-robustness, which refers to unwanted robustness against admissible perturbations with changed semantic content.

However, it is unclear how these adversarial defense techniques can be adapted to the more general setting of hypergraphs. While there are studies on attacks targeting the structure of hypergraphs (e.g., Hu et al. (2023)), there has been no research on adversarial attacks on node features in hypergraphs. This paper reveals that even naturally occurring noise such as Gaussian distributed random feature noise, can significantly impact the predictive performance of hypergraph neural networks. Such Gaussian distributed noise is pervasive in practice, and is known as classical measurement error Carroll et al. (2006). It arises during data collection, labeling, and preprocessing stages and may not be intentionally added by individuals. To our knowledge, no technique exists that aims to improve the denoising capability of state-of-the-art two-stage hypergraph learning models in the presence of node feature noises.

## 3 METHODOLOGY

In this section, we mainly explain our ideas in the context of node classification on hypergraphs. Let $G = (\mathcal{V}, \mathcal{E})$ be a hypergraph, where $\mathcal{V}$ is the set of nodes and $\mathcal{E}$ is the set of hyperedges. Each hyperedge can contain two or more nodes, and let $V(e)$ be the nodes in hyperedge $e$. Two hyperedges are neighbors if they share some common node. The structure of a hypergraph is represented by an incidence matrix $\mathcal{H} \in \{0, 1\}^{|\mathcal{V}| \times |\mathcal{E}|}$, where each entry $\mathcal{H}(v, e)$ indicates whether the node $v$ is in the hyperedge $e$. There can be Gaussian noise $\mathcal{N}(\mu, \sigma^2)$ added to (every dimension of) latent node features $X$ to get observed node features $\tilde{X}$. Our task is to learn a function $\mathcal{F} : \mathcal{V} \to \mathcal{C}$ which maps each node $v \in \mathcal{V}$ with noise feature to one class in $\mathcal{C}$.

**Two-Stage Hypergraph Message-Passing Framework** Before discussing our improvement, we first explain the existing two-stage hypergraph message-passing framework Arya et al. (2020); Huang

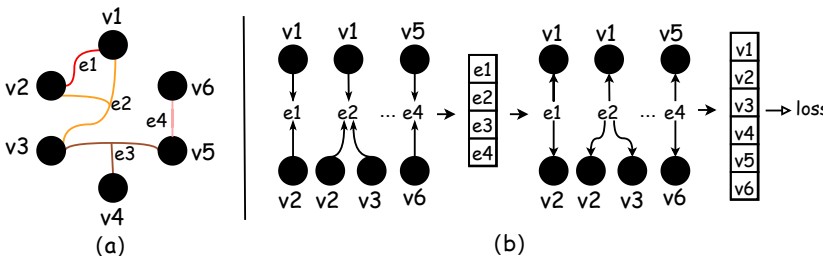

Figure 1: (a) An Input Hypergraph; (b) Two-Stage Hypergraph Message-Passing Framework.

& Yang (2021); Chien et al. (2022); Wang et al. (2023). The first stage is the *node-to-hyperedge aggregation*: features/messages of nodes in the same hyperedge are passed to the hyperedge and aggregated based on a permutation-invariant aggregation function $f_{\mathcal{V} \to \mathcal{E}}$. This can be considered that hyperedge representations are learned *explicitly*. The second stage is the *hyperedge-to-node aggregation*: the updated representation of a hyperedge is passed back to its nodes. Nodes present in multiple hyperedges receive messages from these hyperedges, which are then aggregated through a permutation-invariant function $f_{\mathcal{E} \to \mathcal{V}}$, completing the one-layer message-passing process. The functions $f_{\mathcal{V} \to \mathcal{E}}$ and $f_{\mathcal{E} \to \mathcal{V}}$ can be simply the sum function, the mean function, Multi-Layer Perception (MLP), or recent advances in deep learning of multiset functions Chien et al. (2022). Fig. 1(a) shows an input hypergraph which serves as a running example used throughout the paper. Fig. 1(b) illustrates the two-stage message-passing framework, where features of $v_1$ and $v_2$ are passed to $e_1$, features of $v_1$, $v_2$ and $v_3$ are passed to $e_2$, etc. Then the updated representations of $e_1, \cdots e_4$ are passed back to their nodes for message aggregations and updating of node representations. Formally, the two-stage hypergraph message-passing framework can be summarized as the following two equations:

$$x_e^{(l+1)} = f_{\mathcal{V} \to \mathcal{E}}(x_e^{(l)}, \{x_v^{(l)}\}_{e \ni v}; \Theta^{(l)}) \tag{1}$$

$$x_v^{(l+1)} = f_{\mathcal{E} \to \mathcal{V}}(x_v^{(l)}, \{x_e^{(l)}\}_{v \in e}; \Phi^{(l)}) \tag{2}$$

where $x_e^{(l)}$ and $x_v^{(l)}$ are the latent representations of hyperedges and nodes at layer $l$, respectively, and $\Theta^{(l)}$ and $\Phi^{(l)}$ are the learnable parameters at layer $l$. Although ED-HNN applies specialized hyperedge-to-node messages in Eq. (2), it also falls into the two-stage message-passing framework and suffers from the following problems.

We observe two key shortcomings in the two-stage message-passing. First, there is no communication between neighboring hyperedges within a single layer. That means that a hyperedge's learned representation cannot be immediately passed to its neighboring hyperedge within a layer, hindering fast information aggregation. For example in Fig. 1(a), hyperedges $e_2$ and $e_3$ cannot aggregate each other's features within one layer, even though they are neighbors. Second, unlike traditional graph neural networks where neighboring nodes exchange messages directly, in the two-stage message-passing, neighboring nodes do not directly exchange messages but communicate through shared hyperedges indirectly. For example, neighboring nodes $v_3$ and $v_4$ will not directly pass messages to each other, instead they impact each other through hyperedge $e_3$ in an indirect way. Additionally, the message from $v_4$ to $v_3$ contains complete features of $v_4$, while the message from $e_3$ to $v_3$ only includes the partial information about how $v_4$'s features influence $e_3$. As we will show shortly, direct information exchange between neighboring hyperedges and neighboring nodes in a network layer can help enlarge receptive fields. This not only facilitates information aggregation, but also improves the robustness and denoising capability of hypergraph message-passing models. One might argue that stacking multiple layers of two-stage message-passing can also enlarge the receptive fields. However, neighboring hyperedges and neighboring nodes still do not directly pass messages with multiple layers. Consequently, indirect messages from hyperedges (nodes, respectively) do not contain full features of neighboring nodes (neighboring hyperedges), although the receptive fields become larger. Consistent with this analysis, we find that stacking network layers often leads to worse empirical denoising performance compared with our method of increasing receptive fields. Furthermore, it could suffer from overfitting problem, high training costs, or oversmoothing problem when the underlying method is not designed to counteract oversmoothness. We can combine both methods if needed to further improve robustness and denoising capability, as discussed in Section A.3.

**Receptive Fields** The concept of receptive fields is commonly used in the analysis of Convolutional Neural Networks (CNNs) Luo et al. (2016) and Graph Neural Networks (GNNs) Ma et al. (2021). We now extend this concept to HGNNs and formally define it as follows.

**Definition 1.** *Let the set of nodes that can influence the representation of a node $v$ be $R(v)$. Then the receptive field of $v$ is $|R(v)|$.*

For example, the receptive field of node $v_1$ in a two-stage message-passing layer is $|R(v_1)| = |\{v_1, v_2, v_3\}| = 3$ and the receptive field of $v_5$ is $|R(v_5)| = |\{v_3, v_4, v_5, v_6\}| = 4$.

We now establish the connection between the denoising capability and the receptive fields. Consider that observed node features $\tilde{X}$ is obtained by adding Gaussian noise $\mathcal{N}(\mu, \sigma^2)$ to latent node features $X$. Note that Gaussian distribution is a representative and most common type of noise distributions in practice Guo et al. (2011), which is also known as classical measurement error Carroll et al. (2006). The mean noise $\mu$ is often zero while the standard deviation $\sigma$ determines the magnitude of the noise. The mean noise $\mu$ could be larger than zero, when the feature collection and measurement are carried out in highly noisy settings. Assume $\mu = 0$ to simplify analysis. Let $\tilde{X}_i$ and $X_i$ be the $i^{th}$ entries in $\tilde{X}$ and $X$ respectively. Then we have

$$\tilde{X}_i = X_i + N_i, \text{ where } N_i \sim \mathcal{N}(0, \sigma^2) \tag{3}$$

When the noises $N_i$ are independently and identically distributed (i.i.d.), it is clear that their summation $\sum_{i=1}^{R} N_i$ and the average value $\frac{1}{R} \sum_{i=1}^{R} N_i$ approach to zero as the number $R$ of nodes involved in the summation approaches infinity.

$$\lim_{R \to \infty} \left( \frac{1}{R} \sum_{i=1}^{R} N_i \right) = \mu = 0 \tag{4}$$

This is because the mean noise $\mu$ is assumed to be zero. Intuitively, as the receptive field of a center node becomes larger, the number of nodes involved in the sum or mean function increases. This allows some of the noise may cancel each other out, reducing the overall noise and making it smaller. In practice, the summation of noise may be a weighted sum. For example, in Fig. 1, $v_1$ receives messages from $e_1$ and $e_2$, where $v_2$ is in both of them and its noise has a larger impact/weight than $v_3$'s. Additionally, when there are more than one layer, say two layers, the noise of 1-hop neighbors is added twice, while the noise of 2-hop neighbors is added once. This results in a weighted sum where the noise from 1-hop neighbors is assigned a larger weight. However, we conjecture that with larger receptive fields, it is more likely that appropriate learned weight matrices $\Theta$ and $\Phi$ can adjust the weights in the weighted sum to reduce the overall impact of node feature noises.

**Conjecture 1.** *Consider zero-mean Gaussian noise and the sum or the mean aggregation function. As a center node's receptive field increases, the overall impact of the node feature noise has a higher chance to be smaller, which leads to an enhancement of the denoising capability and robustness.*

Although the analysis only holds for $\mu = 0$, we conduct an empirical evaluation on Gaussian noise with $\mu \neq 0$ and demonstrate the effectiveness and transferability of our enhanced models in this setting. In the next section, we introduce a novel method to address the weaknesses mentioned above, increase receptive fields, and enhance the denoising capability of existing two-stage hypergraph message-passing models.

**Proposed Four-Stage Hypergraph Message-Passing Framework** As indicated earlier, the two problems of the two-stage message-passing framework are the lack of direct communication between neighboring hyperedges and between neighboring nodes within a single layer (and across multiple layers as well). To enhance receptive fields and improve denoising capability and robustness, we propose two new modules: the *Hyperedge Communication* module and the *Node Communication* module. These modules are designed to promote direct, accurate information exchange between neighboring hyperedges and between neighboring nodes. addressing the respective issues of prior two-stage methods. We also suggest inserting the Hyperedge Communication module after the node-to-hyperedge aggregation, and the Node Communication module after the hyperedge-to-node aggregation. The overview of the four-stage hypergraph message-passing framework is shown in Fig. 2 and described below.

1. The node-to-hyperedge aggregation: features/messages of nodes in the same hyperedge are passed to the hyperedge and aggregated based on $f_{\mathcal{V} \to \mathcal{E}}$ in Equation (1).

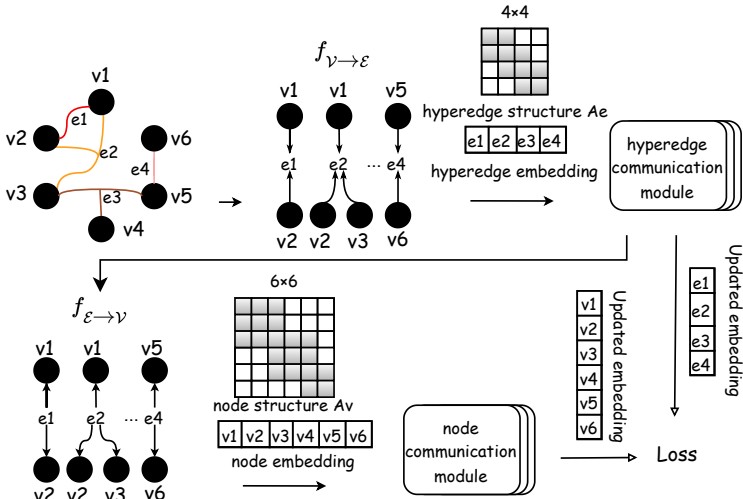

Figure 2: The four-stage hypergraph message-passing framework, enhanced by hyperedge communication and node communication

2. Hyperedge communication: Neighboring hyperedges pass their updated features to each other and further update the hyperedge representation according to Equation (5).

3. The hyperedge-to-node aggregation: Nodes receive the final representations of hyperedges they belong to and update their representations based on $f_{\mathcal{E} \to \mathcal{V}}$ in Eq. (2).

4. Node communication: neighboring nodes exchange their updated node features with each other, and nodes compute their final representations according to Equation (6).

**Hyperedge Communication.** To facilitate direct communication among hyperedges, we define an adjacency matrix on hyperedges, called *hyperedge adjacency matrix* $A_e \in \mathbb{N}^{|\mathcal{E}| \times |\mathcal{E}|}$, as the multiplication of the transpose of incidence matrix $\mathcal{H}$ and $\mathcal{H}$ itself: $A_e = \mathcal{H}^T \mathcal{H}$. Here, each entry $A_e(e_1, e_2)$ indicates *how many common nodes there are* between hyperedges $e_1$ and $e_2$. This allows us to compute the neighbors of a hyperedge $e$ as those hyperedges $e'$ with a non-empty entry $A_e(e, e')$ (except $e$ itself), i.e., those hyperedges sharing at least one common node with $e$. The matrix $A_e$ is very sparse in practice, as will be verified in our experiments in Section A.4 and Table 5. Then sparse matrix multiplication can be employed for all the operations to improve computational efficiency. To remove the effect of hyperedges themselves, we set all the diagonal elements' values to 1. That is, $\hat{A}_e(i,j) = \begin{cases} 1, & i = j \\ A_e(i,j), & i \neq j \end{cases}$ . Considering possibly different ranges of values, we also perform symmetric normalization on $\hat{A}_e$ according to its degree matrix $D_e$. Here, $D_e$ is the diagonal matrix of the edge degrees $d(e) = \sum_{v \in e} \mathcal{H}(v, e)$. Additionally, we add residual connections He et al. (2016) as a means to alleviate the over-smoothing phenomenon. Then the hyperedge communication module can be represented by the following equation:

$$\tilde{x}_e^{(l+1)} = D_e^{-\frac{1}{2}} \hat{A}_e D_e^{-\frac{1}{2}} x_e^{(l+1)} W_e + x_e^{(l+1)}, \tag{5}$$

where $x_e^{(l+1)} \in \mathbb{R}^{|\mathcal{E}| \times F_e}$ is the latent representation of hyperedge $e$ at layer $l + 1$ with feature dimension $F_e$ and $W_e$ is the learnable parameter. In this way, neighboring hyperedges can communicate with each other through their shared common nodes. For example in Fig. 2, hyperedges $e_2$ and $e_3$ now engage in each other's message-passing process because they share a common node $v_3$, effectively enabling a larger receptive field within a single layer.

Eq. (5) follows the principle that the more neighbors a hyperedge has, the more information aggregation occurs, and the larger the receptive fields are. Hyperedge $e_3$ can communicate with both $e_2$ and $e_4$, while $e_1$ can only communicate with $e_2$. Thus, the nodes in $e_3$ enjoy larger receptive fields than nodes in $e_1$. In addition, Eq. (5) considers the connection strength of hyperedges into the message-passing computation. The more common nodes two neighboring hyperedges share, the stronger the connection strength is, and the more weight will be put on the updating. For example, in

Fig. 2, the weight of the message between $e_2$ and $e_1$ is larger than that between $e_2$ and $e_3$, because $e_2$ shares more common nodes with $e_1$ than with $e_3$.

**Node Communication.** Inspired by traditional graph neural networks, we aim to enable direct message passing between neighboring nodes, instead of only indirect communication in the two-stage methods. We define the node adjacency matrix as $A_v \in \mathbb{N}^{|\mathcal{V}| \times |\mathcal{V}|}$ as $A_v = \mathcal{H}\mathcal{H}^T$. Here, each entry $A_v(v_1, v_2)$ indicates *how many hyperedges both nodes $v_1$ and $v_2$ are in*. In this way, we can easily compute neighboring nodes of $v$, i.e., those nodes $v'$ with a non-empty entry $A_v(v, v')$ except $v$ itself, as well as their connection strength. In order to remove the effect of nodes themselves, we also set diagonal elements of $A_v$ as 1 to get $\hat{A}_v$. Then symmetric normalization is performed on $\hat{A}_v$ by its degree matrix $D_v$, where $D_v$ is the diagonal matrix of the node degrees. Similarly, residual connections are added to circumvent oversmoothness. The node communication module can now be represented by the following equation:

$$\tilde{x}_v^{(l+1)} = D_v^{-\frac{1}{2}} \hat{A}_v D_v^{-\frac{1}{2}} x_v^{(l+1)} W_v + x_v^{(l+1)} \tag{6}$$

where $x_v^{(l+1)} \in \mathbb{R}^{|\mathcal{V}| \times F_v}$ is the latent representation of nodes at layer $l + 1$ with feature dimension $F_v$ and $W_v$ is the learnable parameter. Based on Eq. (6), neighboring nodes in a hyperedge can pass their updated representations to each other to compute the final node representation. For example in Fig. 2, nodes $v_1$ and $v_3$ pass their updated features to each other because both are part of hyperedge $e_2$. Remarkably, the more neighboring nodes a node has, the more information aggregation there is towards a larger receptive field. Node $v_3$ can interact with more nodes ($v_1, v_2, v_4, v_5$) and thus has a larger receiptive field than $v_6$, which is in fewer hyperedges.

Moreover, according to Eq. (6), node communication follows the principle that the more parallel hyperedges two nodes share, the larger the weight of their communication is. For example in Fig. 2, the weight of the communication between $v_1$ and $v_2$ is larger than that between $v_1$ and $v_3$ because $v_1$ and $v_2$ share two hyperedges ($e_1$ and $e_2$) and $v_1$ and $v_3$ are only in one hyperedge $e_2$.

As of now, we have completed the definition of a four-stage hypergraph message-passing layer. We intentionally add a skip-connection from hyperedge representations to the loss function as shown in Fig. 2. We can stack multiple network layers and obtain the final representations of nodes and hyperedges. When both nodes and hyperedges have groundtruth labels, we combine them to formulate the objective function as defined below:

$$Loss = -\sum_{v \in \mathcal{V}} \sum_{k \in \mathcal{C}} y_{v,k} \ln(\hat{y}_{v,k}) - \lambda \sum_{e \in \mathcal{E}} \sum_{k \in \mathcal{C}} y_{e,k} \ln(\hat{y}_{e,k}) \tag{7}$$

where the two terms represent the cross-entropy loss for nodes and hyperedges, respectively. Here, $y$ and $\hat{y}$ denote the label values and predicted values, respectively. $\hat{y}_{v,k}$ and $\hat{y}_{e,k}$ are the predicted values for nodes and hyperedges belonging to class $k$ and they can be computed by applying the softmax function to the output of the final layer as shown below:

$$\hat{y}_{*,k} = \frac{x_{*,k}^{(l)}}{\sum_{i=0}^{n} x_{*,i}^{(l)}} \tag{8}$$

In Eq. (7), $\lambda$ is the hyperparameter for adjusting the contribution of the hyperedge loss. In case the hyperedge labels are absent, $\lambda$ is set to 0.

With the insertion of new modules, we have to increase the computational complexity of the two-stage message-passing models. However, this does not imply that the training convergence time must increase. In fact, we observe that the convergence time remains comparable to that of the two-stage counterparts, as will be shown in the experimental section.

**Theoretical Justification** We first provide the theoretical analysis on receptive fields in the two-stage message-passing process. Let $N_v(v) = \{v' : A_v(v, v') > 0, v' \neq v\}$ be the neighbors of a node $v$ in $A_v$. We show that in the two-stage hypergraph message-passing framework, the receptive field $|R(v)|$ is exactly $|N_v(v) \cup \{v\}|$. We then present two lemmas summarizing the improvement in the receptive field after adding each of the developed modules.

**Lemma 1.** *The receptive field of a node $v$ after introducing the hyperedge communication module is $|\cup_{v' \in N_v(v)} (v' \cup N_v(v')) \cup \{v\}|$, which is no smaller than the receptive field, $|N_v(v) \cup \{v\}|$, in the two-stage message-passing framework.*

**Lemma 2.** *The receptive field of a node $v$ after introducing the node communication module is* $|\cup_{v' \in N_v(v)} (\{v'\} \cup N_v(v')) \cup \{v\}|$, *which is no smaller than* $|N_v(v) \cup \{v\}|$.

The two lemmas prove that the theoretical improvement in the receptive field of a node due to the addition of the two new modules is the same. See an example in Fig. **??**. The receptive field of $v_1$ in the two-stage message-passing is $R(v_1) = |\{v_1, v_2\}| = 2$, while after adding either hyperedge communication or node communication, the receptive field becomes $R(v_1) = |\{v_1, v_2, v_3, v_4\}| = 4$.

However, in practice, including both modules as in our four-stage message-passing framework can further enhance the denoising capability and robustness. The supporting evidence and discussions can be found in our ablation study.

We further prove that the four-stage message-passing framework has a larger or the same receptive field of nodes compared to the two-stage method in Theorem 1. In particular, when the 2-hop neighbors of a node are strictly larger than its 1-hop neighbors, its receptive field gets strictly larger. In our experiments, a considerable portion (on average 62%) of nodes' receptive fields experience an increase as discussed in Appendix, Table 6.

**Theorem 1.** *The receptive field of a node $v$ in the four-stage message-passing framework is* $|\cup_{v' \in N_v(v)} (v' \cup N_v(v')) \cup \{v\}|$, *no smaller than the receptive field,* $|N_v(v) \cup \{v\}|$, *in the two-stage message-passing. It is strictly larger when the 2-hop neighbors of $v$ are a proper superset of its 1-hop neighbors. That is,* $\cup_{v' \in N_v(v)}(N_v(v'))/(N_v(v) \cup v) \neq \emptyset$.

## 4 EXPERIMENTS

Here we summarize the results of our extensive experiments with the complete set of results reported in Appendix. Deferred discussions include effects of stacking multiple layers, sparsity and receptive field, and non-zero mean Gaussian, node embedding visualization, and hyperparameter settings.

**Experimental Setups** We apply the proposed four-stage message-passing framework to three representative two-stage models, UniGCNII Huang & Yang (2021), AllSetTransformer Chien et al. (2022), and ED-HNN Wang et al. (2023), resulting in our enhanced models, *EnUniGCNII*, *EnAllSet*, and *EnEDHNN* respectively. We thoroughly compare our three methods against their original counterparts as well as five other representative hypergraph learning methods, including MLP (2 layers with 64 neurons each), HNHN Dong et al. (2020), HGNN Feng et al. (2019), HCHA Bai et al. (2021), and HyperGCN Yadati et al. (2019). The network architectures for these models and hyperparameter settings follow from the source code of AllSetTransformer, as reported in Appendix. We adopt five public benchmark datasets: Cora, Citeseer, Pubmed, Cora-CA, DBLP-CA Yadati et al. (2019) and 20Newsgroups Dua et al. (2017) with statistics reported in Table 2. By default, we add zero-mean Gaussian noise $\mathcal{N}(0, \sigma^2)$ to each dimension of node features in every dataset before training, where Gaussian is the most popular noise distribution in real world and standard deviation $\sigma = 0, 0.6, 1$ are used to simulate different noise levels: *clean, moderate*, and *heavy*. To comprehensively evaluate the quality of the learned node and hyperedge embeddings, we consider both hypernode classification and hyperedge prediction tasks.

**Node Classification** We choose accuracy as the evaluation metric for node classification, running 20 runs, and reporting the mean accuracy along with standard deviation in Table 1. When heavy noise ($\sigma = 1$) is present, all existing methods suffer significant accuracy degradation compared to the clean setup ($\sigma = 0$). For example, the average performance of ED-HNN, UniGCNII and HyperGCN decrease by 24%, 29.6% and 45.9%, respectively. This verifies that existing methods are not robust to Gaussian noise unfortunately. Our models EnUniGCNII, EnAllSet and EnEDHNN greatly alleviate this situation. EnUniGCNII attains higher accuracy than its original counterpart UniGCNII by about 11% on the Cora dataset, 7% on DBLP-CA dataset, and an average of 5% across all datasets. EnAllSet outperforms the original counterpart AllSetTransformer by about 5% on the Cora dataset, 4% on Citeseer, and an average of 2.1%. EnEDHNN outperforms the original counterpart ED-HNN by about 5% on the Cora dataset, 4% on Cora-CA, and an average of 3%.

Moreover, our enhanced models always attain the best denoising performance in almost all datasets, except for 20Newsgroups. Note that this dataset is the only one with a small feature dimension of only 100. The added continuous Gaussian noise does not significantly impact AllSetTransformer and ED-HNN, thereby limiting their room for improvement. Remarkably, in the clean setup with $\sigma = 0$, our three models maintain comparable test accuracy to their original counterparts. This property supports the preference for our models, as users may not know the noise setup of an input hypergraph in practice.

Table 1: Experimental results for node classification (accuracy with standard deviation in %).

| | Cora | | | Citeseer | | | Pubmed | | |
|---|---|---|---|---|---|---|---|---|---|
| $\mathcal{N}(0,\sigma^2)$ | 1 | 0.6 | 0 | 1 | 0.6 | 0 | 1 | 0.6 | 0 |
| MLP | 17.37±1.17 | 20.64±1.51 | 68.37±1.48 | 18.89±1.37 | 21.36±1.38 | 69.71±1.39 | 37.62±0.45 | 38.38±0.65 | 75.77±0.85 |
| HNHN | 32.50±2.17 | 35.90±1.85 | 76.06±1.42 | 27.91±2.14 | 31.73±1.76 | 73.03±1.40 | 41.48±0.97 | 41.41±0.82 | 85.57±0.51 |
| HGNN | 48.32±1.72 | 51.94±1.41 | 78.52±1.35 | 37.90±1.71 | 40.67±1.54 | 72.01±1.30 | 45.37±0.60 | 44.77±0.67 | 85.52±0.50 |
| HCHA | 48.77±1.59 | 51.13±1.32 | 79.01±1.90 | 38.70±1.61 | 40.47±2.23 | 72.44±1.14 | 44.90±0.56 | 44.50±0.70 | 85.74±0.51 |
| HyperGCN | 25.92±1.88 | 30.79±1.78 | 73.12±1.30 | 21.67±1.10 | 23.97±1.79 | 65.07±1.43 | 31.69±0.63 | 31.55±1.08 | 74.18±7.32 |
| UniGCNII | 39.51±1.77 | 44.05±2.07 | 78.28±1.90 | 33.36±1.83 | 38.41±1.89 | 71.83±2.19 | 44.69±0.71 | 44.56±0.71 | 88.36±0.46 |
| AllSetTransformer | 46.81±1.96 | 50.16±1.95 | 79.18±1.39 | 35.72±1.85 | 39.79±1.15 | 72.15±1.74 | 44.53±0.92 | 44.92±0.71 | 88.48±0.54 |
| ED-HNN | 45.72±1.26 | 50.20±1.52 | **79.92**±**1.24** | 34.89±1.96 | 39.39±1.62 | 73.91±1.34 | 44.99±0.91 | 45.78±0.69 | **88.57**±**0.39** |
| *EnUniGCNII (Ours)* | 50.24±1.85 | 53.15±2.16 | 78.66±1.43 | **39.75**±**2.17** | 42.37±2.06 | 70.28±1.05 | 46.03±0.47 | 46.64±0.65 | 87.60±0.45 |
| *EnAllSet (Ours)* | **51.03**±**1.83** | **53.43**±**1.72** | 79.27±1.52 | 39.36±2.09 | 41.38±0.96 | 72.25±1.53 | 45.25±0.95 | 45.41±0.74 | 88.14±0.33 |
| *EnEDHNN (Ours)* | 50.46±1.90 | 53.41±2.02 | 79.77±1.47 | 39.71±1.67 | **42.72**±**1.56** | 72.65±1.11 | **46.73**±**0.59** | **46.85**±**0.82** | 88.02±0.44 |
| | Cora-CA | | | DBLP-CA | | | 20Newsgroups | | |
| $\mathcal{N}(0,\sigma^2)$ | 1 | 0.6 | 0 | 1 | 0.6 | 0 | 1 | 0.6 | 0 |
| MLP | 17.21±1.57 | 21.23±1.77 | 68.06±1.63 | 37.01±0.30 | 49.68±0.35 | 83.40±0.35 | 40.05±0.81 | 47.98±0.74 | 79.03±0.53 |
| HNHN | 31.06±2.00 | 33.94±1.71 | 77.74±0.99 | 45.61±0.43 | 57.71±0.42 | 86.71±0.33 | 39.81±0.78 | 47.92±0.64 | 81.38±0.62 |
| HGNN | 67.91±2.15 | 69.42±1.77 | 82.29±1.50 | 79.51±0.36 | 82.23±0.49 | 90.96±0.22 | 73.06±0.71 | 76.80±0.57 | 80.75±0.71 |
| HCHA | 67.62±1.62 | 70.04±1.41 | 82.84±0.76 | 79.43±0.38 | 82.21±0.34 | 90.87±0.21 | 73.59±0.82 | 76.84±0.74 | 80.85±0.59 |
| HyperGCN | 25.64±1.78 | 31.42±1.32 | 75.20±1.83 | 35.40±1.21 | 53.31±0.62 | 86.32±0.34 | 35.71±1.17 | 45.05±0.92 | 77.62±0.77 |
| UniGCNII | 58.59±1.85 | 62.42±1.86 | **84.01**±**1.67** | 79.39±0.38 | 83.19±0.44 | 91.67±0.22 | 62.22±0.64 | 68.39±0.73 | 81.09±0.64 |
| AllSetTransformer | 66.78±1.35 | 68.86±1.53 | 83.32±1.42 | 83.93±0.24 | 86.12±0.38 | 91.49±0.28 | **80.56**±**0.54** | **80.75**±**0.54** | **81.60**±**0.60** |
| ED-HNN | 65.33±1.69 | 68.38±1.66 | 83.94±1.17 | 83.78±0.43 | 86.68±0.32 | **91.88**±**0.18** | 77.82±0.73 | 79.13±0.66 | 81.15±0.69 |
| *EnUniGCNII (Ours)* | 69.51±2.33 | **71.07**±**1.71** | 82.17±1.06 | **86.58**±**0.38** | **87.93**±**0.24** | 91.27±0.23 | 73.51±0.72 | 75.85±0.72 | 80.14±0.65 |
| *EnAllSet (Ours)* | **69.67**±**1.45** | 70.98±2.59 | 83.71±1.44 | 85.81±0.29 | 87.15±0.38 | 91.61±0.29 | 80.03±0.39 | 79.36±0.54 | 81.31±0.64 |
| *EnEDHNN (Ours)* | 69.19±1.50 | 70.18±1.33 | 81.38±1.25 | 86.49±0.25 | 87.79±0.32 | 91.76±0.15 | 78.01±0.60 | 79.77±0.47 | 80.88±0.58 |

Comparing our three models themselves, they have comparable performance without a clear winner. EnEDHNN seems to enjoy slightly higher and more stable test accuracy across all settings. When the noise level is reduced to moderate with standard deviation $\sigma = 0.6$, existing methods still experience performance drops, but the reductions in test accuracy are smaller due to less noise. Our three models can still greatly boost the performance of UniGCNII, AllSetTransformer and ED-HNN, achieving state-of-the-art results.

An extra set of experiments on *non-zero mean* Gaussian noise with $\mu = [0, 1, 2, 3]$ and $\sigma = [0, 0.6, 1]$ were conducted with the results plotted in Figures 5, 6, and 7 in Appendix. It is clear to see drastic performance boost across a wide range of noise settings. The proposed four-stage methods show improved denoising capability and excellent transferability for non-zero Gaussian noise.

**Ablation Study** We conduct ablation experiments on our proposed models and report the results in Figures 8, 9, and 10 in Appendix. We refer to the initial two-stage model as *None*, the model with only hyperedge communication module as *E*, with only node communication module as *N*, and the full model as *EN*. It is clear that the combination of both modules consistently outperforms using any single module (only *E* or only *N*), which further significantly improves *None*.

**Convergence Time Analysis** We plot the *loss vs. time curves* on the DBLP-CA dataset (the largest tested dataset) under noise standard deviation $\sigma = 0, 1$ in Figures 3 and 4 in Appendix. The results show that the convergence time of our models and their original counterparts are very close to each other, with a difference smaller than 2 seconds, which is generally acceptable. Hence, our method of equipping denoising capability in HGNNs does not introduce overwhelming computational overhead. Additionally, we find that our hyperedge communication and node communication modules can help the model reduce instability in the loss function. For example in UniGCNII, its loss often heavily fluctuates during the convergence process, while this issue is effectively addressed in EnUniGCNII.

**Hyperedge Prediction** The hyperedge prediction task aims to determine whether there exists a hyperedge among nodes in a given set. We compute the hyperedge embedding as the sum of nodes' embedding for all nodes within a hyperedge, followed by MLP for binary classification learning. The results in Table 7 of Appendix verify the excellent performance of our models, particularly EnEDHNN, which achieves the highest accuracy with a large gap over the second-best model. Compared to their original counterparts, EnUniGCNII and EnEDHNN achieve average performance improvements of 9% and 3% respectively. As expected, when noise is introduced, the accuracy of almost all models deteriorates. However, we observe the performance degradation is less severe than in node classification, ranging from a few percent to 12%. This indicates that node feature noise has a smaller impact on hyperedge prediction compared to node classification. This aligns well with previous observations that structural changes affect edge prediction accuracy more significantly than node classification Yang et al. (2020).

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

# A    ADDITIONAL EXPERIMENTS

## A.1    EXPERIMENTAL SETUPS

We apply the proposed four-stage message-passing framework to three representative two-stage models, UniGCNII Huang & Yang (2021), AllSetTransformer Chien et al. (2022), and ED-HNN Wang et al. (2023), resulting in our enhanced models, *EnUniGCNII*, *EnAllSet*, and *EnEDHNN* respectively. We thoroughly compare our three methods against their original counterparts as well as five other representative hypergraph learning methods, including MLP (2 layers with 64 neurons each), HNHN Dong et al. (2020), HGNN Feng et al. (2019), HCHA Bai et al. (2021) and HyperGCN Yadati et al. (2019). The network architectures for these models and hyperparameter settings follow from the source code provided by AllSetTransformer Chien et al. (2022), e.g., the number of layers is one for all models except MLP. The MLP considers each instance (hypernode) as an independent and identically distributed (i.i.d) sample, without exploiting the hypergraph structure for making predictions. Our code is available at https://anonymous.4open.science/r/FourStageHGNN-59AB/

## A.2    DATASETS

To facilitate reproducibility, we use the following public benchmarking datasets: Cora, Citeseer, Pubmed, Cora-CA, DBLP-CA Yadati et al. (2019) and 20Newsgroups Dua et al. (2017). In Cora, Citeseer and Pubmed, one document represents a node and all documents cited by a document are connected by a hyperedge. We call them co-citation data. In Cora-CA and DBLP-CA, one document represents a node and all documents co-authored by an author are in one hyperedge. We call them co-authorship data. In the 20Newsgroups dataset, one new represents a node and all news sharing a common attribute are in the same hyperedge.See Table 2 for the full statistics of all tested datasets in our experiments. To comprehensively evaluate the quality of the learned node and hyperedge embeddings, we consider both hypernode classification and hyperedge prediction tasks.

To fairly evaluate the performance of all tested methods, we align our experimental settings with AllSetTransformer Chien et al. (2022). We randomly divide the data into train/valid/test sets with a

Table 2: Dataset statistics information

| DATA SET | $|\mathcal{V}|$ | $|\mathcal{E}|$ | FEATURE | CLASS | MAX $|e|$ | MIN $|e|$ | AVG $|e|$ |
|---|---|---|---|---|---|---|---|
| CORA | 2708 | 1579 | 1433 | 7 | 5 | 2 | 3.03 |
| CITESEER | 3312 | 1079 | 3703 | 6 | 26 | 2 | 3.2 |
| PUBMED | 19717 | 7963 | 500 | 3 | 171 | 2 | 4.35 |
| CORA-CA | 2708 | 1072 | 1433 | 7 | 43 | 2 | 4.28 |
| DBLP-CA | 41302 | 22363 | 1425 | 6 | 202 | 2 | 4.45 |
| 20NEWSGROUPS | 16242 | 100 | 100 | 4 | 2241 | 29 | 654.51 |

50%/25%/25% split and aggregate results from 20 runs using random splits and initializations. To validate model robustness against noise, we add default zero-mean Gaussian noise $\mathcal{N}(0, \sigma^2)$ to each dimension of node features in every dataset before training, where Gaussian is the most popular noise distribution in real world and standard deviation $\sigma = 0, 0.6, 1$ are used to simulate different noise levels: clean/slight, moderate, and heavy. We use the Adam optimizer with a learning rate of 0.001 and fix the training epochs at 500. All experiments were conducted on a server with an Intel E5-2650 CPU (2.20GHz), 256G RAM, and an NVIDIA 1080Ti GPU (11GB).

**Node Classification** We choose accuracy as the evaluation metric for node classfication, running 20 runs, and reporting the mean accuracy along with standard deviation in Table 1. When heavy noise ($\sigma = 1$) is present, all existing methods suffer significant accuracy degradation compared to the clean setup ($\sigma = 0$). For example, the average performance of ED-HNN, UniGCNII and HyperGCN decrease by 24%, 29.6% and 45.9%, respectively. This verifies that existing methods are not robust to Gaussian noise unfortunately. Our models EnUniGCNII, EnAllSet and EnEDHNN greatly alleviate this situation. EnUniGCNII attains higher accuracy than its original counterpart UniGCNII by about 11% on the Cora dataset, 7% on DBLP-CA dataset, and an average of 5% across all datasets. EnAllSet outperforms the original counterpart AllSetTransformer by about 5% on the Cora dataset, 4% on Citeseer, and an average of 2.1%. EnEDHNN outperforms the original counterpart ED-HNN by about 5% on the Cora dataset, 4% on Cora-CA, and an average of 3%.

Moreover, our enhanced models always attain the best denoising performance in almost all datasets, except for 20Newsgroups. Note that this dataset is the only one with a small feature dimension of only 100. The added continuous Gaussian noise does not significantly impact AllSetTransformer and ED-HNN, thereby limiting their room for improvement. Remarkably, in the clean setup with $\sigma = 0$, our three models maintain comparable test accuracy to their original counterparts. This property supports the preference for our models, as users may not know the noise setup of an input hypergraph in practice.

Comparing our three models themselves, they have comparable performance without a clear winner. EnEDHNN seems to enjoy slightly higher and more stable test accuracy across all settings. When the noise level is reduced to moderate with standard deviation $\sigma = 0.6$, existing methods still experience performance drops, but the reductions in test accuracy are smaller due to less noise. Our three models can still greatly boost the performance of UniGCNII, AllSetTransformer and ED-HNN, achieving state-of-the-art results.

By analyzing other representative hypergraph learning methods, we find that the receptive field of a HyperGCN convolution layer is the smallest because it only selects two special nodes within a hyperedge for convolution. Therefore, its denoising capability is always very weak under different noise level in diverse datasets, which aligns with our conjecture on the connection to the receptive fields discussed in Section 3. On the other hand, HGNN and HCHA connect all pairs of nodes within a hyperedge through clique expansion and then utilize GNN message-passing. Although this clique expansion may loss certain higher-order connectivity, leading to suboptimal performance in clean hypergraphs, we find that their robustness is not poor but still clearly worse than ours.

### A.3 EFFECTS OF STACKING MULTIPLE LAYERS

Stacking network layers can increase the receptive fields, and according to our conjecture, this can also improve the denoising capability. To verify this hypothesis, we conduct an experiment by

Table 3: Comparison of multiple layers of AllSetTransformer and UniGCNII and our *one-layer* models. OOM means the Out-Of-Memory error.

| | Cora | | | Citeseer | | |
|---|---|---|---|---|---|---|
| $\mathcal{N}(0,\sigma^2)$ | 1 | 0.6 | 0 | 1 | 0.6 | 0 |
| AllSetTransformer | 46.81±1.96 | 50.16±1.95 | 79.18±1.39 | 35.72±1.85 | 39.79±1.15 | 72.15±1.74 |
| AllSetTransformer 3L | 48.98±1.78 | 51.84±1.89 | 75.78±1.96 | 37.55±2.36 | 38.89±1.39 | 59.73±6.29 |
| AllSetTransformer 6L | 32.84±3.27 | 32.57±3.06 | 39.25±3.20 | 20.51±1.53 | 20.46±1.38 | 33.35±1.29 |
| AllSetTransformer 9L | 30.55±1.80 | 30.59±1.78 | 30.46±1.37 | 20.28±1.60 | 20.27±1.51 | 32.95±2.59 |
| UniGCNII | 39.51±1.77 | 44.05±2.07 | 78.28±1.90 | 33.36±1.83 | 38.41±1.89 | 71.83±2.19 |
| UniGCNII 3L | 47.63±1.73 | 51.53±1.65 | 79.16±1.48 | 37.88±2.37 | 41.50±1.92 | 72.36±1.82 |
| UniGCNII 6L | 49.19±2.09 | 51.08±1.83 | 79.55±1.47 | 38.50±1.63 | 42.03±1.39 | **73.04±1.35** |
| UniGCNII 9L | 47.41±1.67 | 50.81±1.64 | 79.62±1.55 | 38.53±2.05 | 41.24±1.61 | 72.83±1.40 |
| EnUniGCNII (Ours) | 50.24±1.85 | 53.15±2.16 | 78.66±1.43 | **39.75±2.17** | 42.37±2.06 | 70.28±1.05 |
| EnAllSet (Ours) | **51.03±1.83** | **53.43±1.72** | 79.27±1.52 | 39.36±2.09 | 41.38±0.96 | 72.25±1.53 |
| EnEDHNN (Ours) | 50.46±1.90 | 53.41±2.02 | **79.77±1.47** | 39.71±1.67 | **42.72±1.56** | 72.65±1.11 |
| | Pubmed | | | Cora-CA | | |
| $\mathcal{N}(0,\sigma^2)$ | 1 | 0.6 | 0 | 1 | 0.6 | 0 |
| AllSetTransformer | 44.53±0.92 | 44.92±0.71 | **88.48±0.54** | 66.78±1.35 | 68.86±1.53 | 83.32±1.42 |
| AllSetTransformer 3L | 45.87±0.60 | 43.08±2.10 | 88.37±0.42 | 68.76±1.48 | 69.54±1.89 | 80.96±1.27 |
| AllSetTransformer 6L | 39.90±1.35 | 39.80±0.69 | 53.49±7.86 | 39.93±4.36 | 38.00±2.91 | 30.41±1.63 |
| AllSetTransformer 9L | 39.37±0.76 | 39.80±0.77 | 40.04±1.24 | 30.58±1.45 | 30.42±2.01 | 30.89±1.56 |
| UniGCNII | 44.69±0.71 | 44.56±0.71 | 88.36±0.46 | 58.59±1.85 | 62.42±1.86 | 84.01±1.67 |
| UniGCNII 3L | 46.20±0.74 | 46.67±0.44 | 87.83±0.56 | 67.74±1.52 | 69.37±1.79 | 84.15±1.70 |
| UniGCNII 6L | 46.11±0.78 | 46.55±0.75 | 87.78±0.37 | 67.64±1.54 | 69.09±1.58 | **84.34±0.84** |
| UniGCNII 9L | 45.83±0.80 | 46.37±0.68 | 87.58±0.34 | 67.77±1.38 | 69.87±1.68 | 83.94±1.30 |
| EnUniGCNII (Ours) | 46.03±0.47 | 46.64±0.65 | 87.60±0.45 | 69.51±2.33 | **71.07±1.71** | 82.17±1.06 |
| EnAllSet (Ours) | 45.25±0.95 | 45.41±0.74 | 88.14±0.33 | **69.67±1.45** | 70.98±2.59 | 83.71±1.44 |
| EnEDHNN (Ours) | **46.73±0.59** | **46.85±0.82** | 88.02±0.44 | 69.19±1.50 | 70.18±1.33 | 81.38±1.25 |
| | DBLP-CA | | | 20Newsgroups | | |
| $\mathcal{N}(0,\sigma^2)$ | 1 | 0.6 | 0 | 1 | 0.6 | 0 |
| AllSetTransformer | 83.93±0.24 | 86.12±0.38 | 91.49±0.28 | **80.56±0.54** | **80.75±0.54** | **81.60±0.60** |
| AllSetTransformer 3L | OOM | OOM | 91.07±0.21 | 79.58±0.53 | 79.67±0.62 | 80.51±0.59 |
| AllSetTransformer 6L | OOM | OOM | 90.70±0.34 | 39.16±5.80 | 40.83±6.76 | 42.55±7.40 |
| AllSetTransformer 9L | OOM | OOM | 51.60±3.53 | 33.68±0.65 | 33.70±0.64 | 33.74±0.47 |
| UniGCNII | 79.39±0.38 | 83.19±0.44 | 91.67±0.22 | 62.22±0.64 | 68.39±0.73 | 81.09±0.64v |
| UniGCNII 3L | 85.92±0.30 | 87.52±0.35 | **91.95±0.26** | 71.44±0.77 | 74.22±0.61 | 80.90±0.61 |
| UniGCNII 6L | 86.58±0.33 | **87.95±0.29** | 91.89±0.28 | 59.60±0.95 | 66.87±0.83 | 80.95±0.52 |
| UniGCNII 9L | 79.74±0.41 | 83.16±0.4 | 91.73±0.19 | 61.88±0.80 | 68.21±0.63 | 81.06±0.47 |
| EnUniGCNII (Ours) | **86.58±0.38** | 87.93±0.24 | 91.27±0.23 | 73.51±0.72 | 75.85±0.72 | 80.14±0.65 |
| EnAllSet (Ours) | 85.81±0.29 | 87.15±0.38 | 91.61±0.29 | 80.03±0.39 | 79.36±0.54 | 81.31±0.64 |
| EnEDHNN (Ours) | 86.49±0.25 | 87.79±0.32 | 91.76±0.15 | 78.01±0.60 | 79.77±0.47 | 80.88±0.58 |

Table 4: Comparison of multi-layer ED-HNN and multi-layer EnEDHNN. $X$L indicates stacking $X$ layers.

| | Cora | | | Citeseer | | | Pubmed | | |
|---|---|---|---|---|---|---|---|---|---|
| $\mathcal{N}(0,\sigma^2)$ | 1 | 0.6 | 0 | 1 | 0.6 | 0 | 1 | 0.6 | 0 |
| ED-HNN | 45.72±1.26 | 50.20±1.52 | 79.92±1.24 | 34.89±1.96 | 39.39±1.62 | **73.91±1.34** | 44.99±0.91 | 45.78±0.69 | **88.57±0.39** |
| ED-HNN 2L | 49.73±1.84 | 52.80±1.70 | **80.24±1.49** | 36.70±1.07 | 40.95±1.62 | 73.69±1.85 | 45.46±0.83 | 46.16±0.83 | 83.53±12.78 |
| ED-HNN 3L | 50.95±1.71 | 53.00±1.81 | 79.14±1.45 | 38.26±1.83 | 41.92±1.41 | 71.92±1.34 | 46.06±0.64 | 45.90±0.98 | 74.83±20.02 |
| ED-HNN 6L | 50.58±1.88 | **54.19±1.72** | 78.04±1.66 | 39.24±1.57 | 41.38±1.49 | 71.14±1.38 | 46.14±0.88 | 43.66±2.56 | 45.08±6.62 |
| ED-HNN 9L | 51.12±1.74 | 53.43±1.57 | 77.54±2.00 | 38.67±1.48 | 41.97±1.23 | 70.33±1.27 | 45.09±1.02 | 40.48±1.43 | 43.61±2.09 |
| EnEDHNN (Ours) | 50.46±1.90 | 53.41±2.02 | 79.77±1.47 | 39.71±1.67 | 42.72±1.56 | 72.65±1.11 | 46.73±0.59 | **46.85±0.82** | 88.02±0.44 |
| EnEDHNN 2L (Ours) | 51.32±2.00 | 53.94±1.90 | 79.33±1.31 | **40.74±1.42** | **42.93±1.93** | 72.05±1.11 | 46.32±0.84 | 46.84±0.84 | 79.75±16.31 |
| EnEDHNN 3L (Ours) | 51.80±1.86 | 53.69±1.71 | 78.66±1.59 | 39.86±1.77 | 42.67±1.99 | 71.84±1.20 | **46.89±0.51** | 45.55±1.70 | 69.73±1.29 |
| EnEDHNN 6L (Ours) | **52.23±1.59** | 53.73±1.66 | 77.35±1.81 | 39.76±1.73 | 42.30±1.86 | 70.16±1.15 | 45.25±0.87 | 40.97±1.62 | 43.65±6.37 |

| | Cora-CA | | | DBLP-CA | | | 20Newsgroups | | |
|---|---|---|---|---|---|---|---|---|---|
| $\mathcal{N}(0,\sigma^2)$ | 1 | 0.6 | 0 | 1 | 0.6 | 0 | 1 | 0.6 | 0 |
| ED-HNN | 65.33±1.69 | 68.38±1.66 | **83.94±1.17** | 83.78±0.43 | 86.68±0.32 | **91.88±0.18** | 77.82±0.73 | 79.13±0.66 | **81.15±0.69** |
| ED-HNN 2L | 67.90±1.83 | 70.37±2.14 | 82.90±1.26 | 86.41±0.34 | 88.02±0.31 | 91.76±0.23 | 75.74±1.84 | 77.63±0.95 | 80.70±0.63 |
| ED-HNN 3L | 68.28±1.45 | 69.53±1.87 | 82.47±1.22 | 86.93±0.34 | 88.07±0.39 | 91.82±0.26 | 75.49±1.90 | 77.54±1.33 | 80.76±0.67 |
| ED-HNN 6L | 67.89±1.75 | 69.45±1.86 | 81.34±1.18 | 87.13±0.25 | **88.17±0.46** | 91.70±0.23 | 68.33±5.88 | 74.66±2.92 | 80.25±0.58 |
| ED-HNN 9L | 69.22±1.80 | 70.20±1.44 | 59.95±14.64 | 86.57±0.86 | 88.13±0.39 | 91.77±0.27 | 60.33±5.69 | 68.51±6.54 | 79.82±0.63 |
| EnEDHNN (Ours) | 69.19±1.50 | 70.18±1.33 | 81.38±1.25 | 86.49±0.25 | 87.79±0.32 | 91.76±0.15 | **78.01±0.60** | **79.77±0.47** | 80.88±0.58 |
| EnEDHNN 2L (Ours) | 69.46±1.74 | 70.10±1.84 | 81.64±1.78 | 87.00±0.44 | 88.15±0.34 | 91.70±0.23 | 74.28±1.44 | 76.96±1.19 | 80.83±0.70 |
| EnEDHNN 3L (Ours) | 69.73±1.29 | **70.71±1.47** | 81.05±1.26 | 87.10±0.19 | 88.15±0.29 | 91.62±0.33 | 73.28±2.04 | 75.40±2.73 | 80.90±0.57 |
| EnEDHNN 6L (Ours) | **70.06±1.63** | 70.13±1.63 | 81.43±1.40 | **87.28±0.29** | 88.08±0.23 | 91.62±0.26 | 67.38±5.51 | 74.11±2.03 | 80.25±0.49 |

stacking multiple layers of ED-HNN, EnEDHNN, AllSetTransformer, and UniGCNII, studying their impact on node classification accuracy, as shown in Tables 3 and 4. We find that, generally, stacking a few network layers can help boost the denoising capability of the four underlying hypergraph message passing models, which aligns with our conjecture. However, this approach may not always be effective due to increasing model complexity, which can lead to overfitting and worse test accuracy. Methods not designed to counteract oversmoothing, such as AllSetTransformer, also suffer from this problem. One can observe a clear trend of accuracy degradation for multi-layer AllSetTransformer when the number of layers is large in Table 3. When tested on larger datasets (such as DBLP-CA), multi-layer AllSetTransformer easily encounters Out-Of-Memory errors and cannot be used. In contrast, our enhanced models can run successfully on the largest tested dataset, DBLP-CA. Because of all these reasons, our approach to enhancing denoising capability has broader applicability.

Furthermore, in Table 4, we observe that multiple layers of EnEDHNN exhibit stronger denoising performance compared to multiple layers of ED-HNN, winning in 10 out of 12 heavy and moderate noise settings. Our one-layer models already produce superior performance compared to multi-layer AllSetTransformer and multi-layer UniGCNII, as shown in Table 3.

## A.4 SPARSITY AND RECEPTIVE FIELDS

We perform an experiment to assess the sparsity of hyperedge adjacency matrices across all the datasets. The percentage of non-zero entries in a hyperedge adjacency matrix $A_e$ ranges from $9.5 \times 10^{-5}$ to $1.5 \times 10^{-3}$, as shown in Table 5. This allows us to employ sparse matrix data structures, such as sparse matrix multiplication, to significantly improve memory and computational efficiency.

Table 5: Sparsity information of hyperedge adjacency matrices

| Data set | Non-Zero Entries | Zero Entries | Total | Sparsity |
|---|---|---|---|---|
| Cora | 70K | 18M | 18M | 0.0038 |
| Citeseer | 29K | 19M | 19M | 0.0015 |
| Pubmed | 518K | 766M | 766M | 0.00067 |
| Cora-CA | 18K | 14M | 14M | 0.0012 |
| DBLP-CA | 387K | 4,052M | 4,053M | 0.000095 |
| 20Newsgroups | 156K | 267M | 267M | 0.00058 |

In the four-stage message-passing framework, the receptive field of a node $v$ is strictly larger than in the two-stage framework when the set of 2-hop neighbors of $v$ is strictly larger than its set of

Table 6: Percentage of nodes with a larger receptive field in our methods

| Data set | Nodes | Nodes with a larger receptive field | Percentage |
|---|---|---|---|
| Cora | 2708 | 1383 | 51% |
| Citeseer | 3312 | 1162 | 35% |
| Pubmed | 19717 | 3824 | 19% |
| Cora-CA | 2708 | 1967 | 72% |
| DBLP-CA | 41302 | 37820 | 91% |
| 20Newsgroups | 16242 | 16242 | 100% |

1-hop neighbors. Formally, $\cup_{v' \in N_v(v)}(N_v(v'))/(N_v(v) \cup v) \neq \emptyset$. We conduct a quantitative analysis to check whether this equation holds for every node in the datasets under test, with the results presented in Table 6. We observe that a significant majority of the nodes meet this requirement, resulting in a larger receptive field compared to the two-stage message-passing. In summary, although the theoretical analysis only guarantees that the receptive field does not decrease, in practice, a considerable portion (on average 62%) of receptive fields experience an increase.

## A.5 CONVERGENCE TIME ANALYSIS

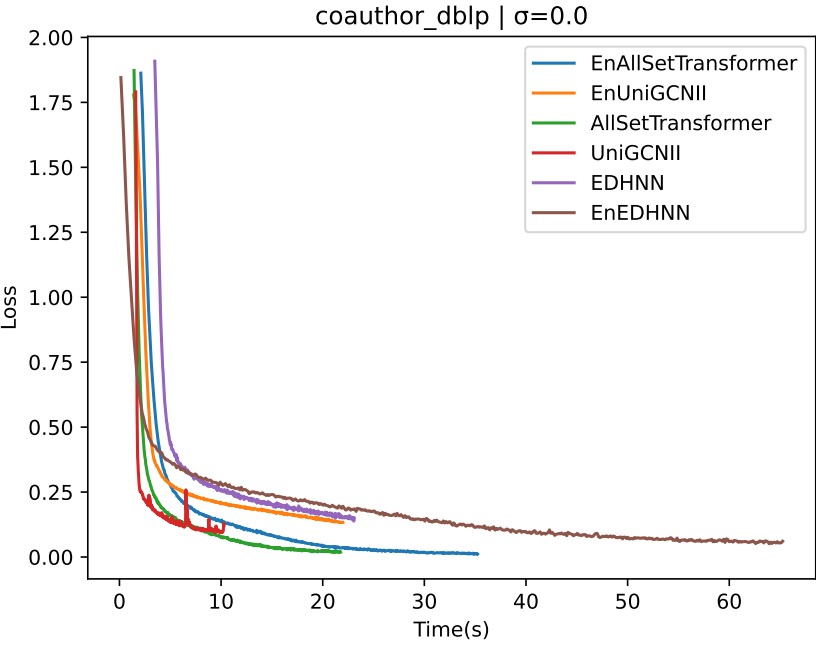

Figure 3: DBLP-CA with noise $\mathcal{N}(0,0)$

We plot the **loss vs. time curves** on the DBLP-CA dataset under noise standard deviation $\sigma = 0, 1$ in Figures 3 and 4, respectively. DBLP-CA is the largest dataset in our experiments, making it valuable as a benchmark for our runtime analysis. Since all tested models converge early within the predefined 500 epochs, we decide to focus on comparing the convergence time among various models instead of the overall training time. One could reduce the maximum number of epochs or adopt an early stopping strategy to save training time.

The results show that the convergence time of EnAllSet, EnUniGCNII and EnEDHNN and their original counterparts are very close to each other, with a difference smaller than 2 seconds, which is generally acceptable. Hence, our method of equipping denoising capability in HGNNs does not introduce overwhelming computational overhead.

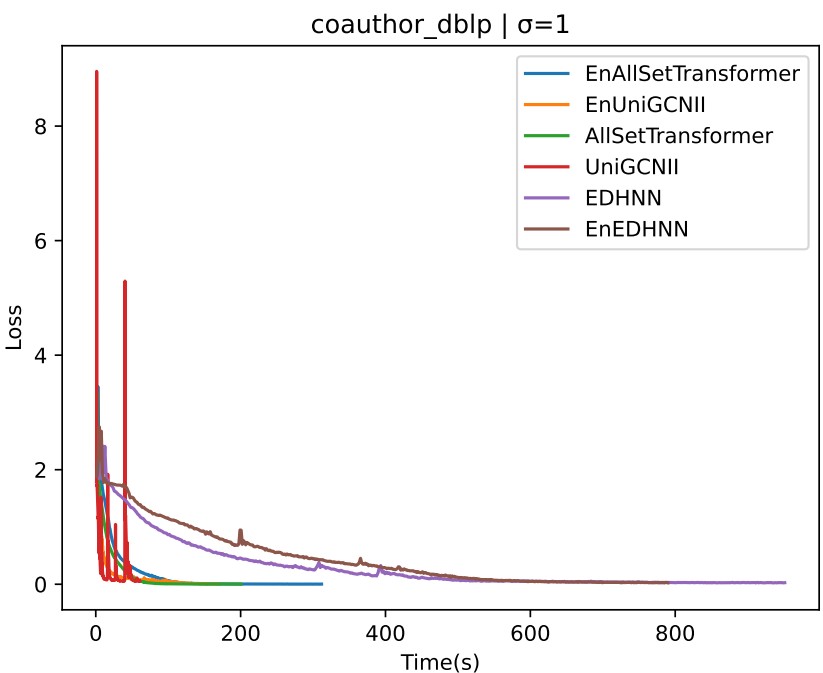

Figure 4: DBLP-CA with noise $\mathcal{N}(0, 1)$

Additionally, we find that our hyperedge communication and node communication modules can help the model reduce instability in the loss function. For example in UniGCNII, its loss often heavily fluctuates during the convergence process, while this issue is effectively addressed in EnUniGCNII.

## A.6 EFFECTS OF NON-ZERO MEAN GAUSSIAN NOISE

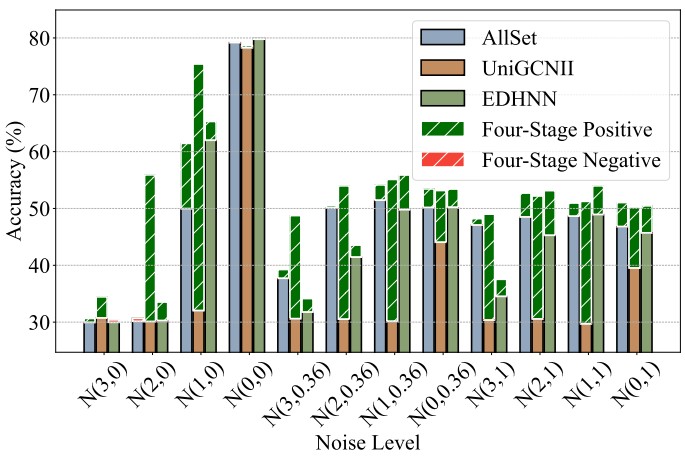

Figure 5: Performance of non-zero mean Gaussian noise in Cora

Although our analysis on receptive fields is only for zero-mean Gaussian noise, we conduct additional experiments on non-zero mean Gaussian noise with mean values of [0, 1, 2, 3] and standard deviation [0, 0.6, 1] in the Cora, Citeseer, and Pubmed datasets with the results plotted in Figures 5, 6, and 7 respectively. In the figures, accuracy improvement or decrease with respect to a two-stage model are outlined in green and red, respectively. It is clear to see drastic performance boost across a wide range of noise settings. The proposed four-stage methods show improved denoising capability and excellent transferability for non-zero Gaussian noise.

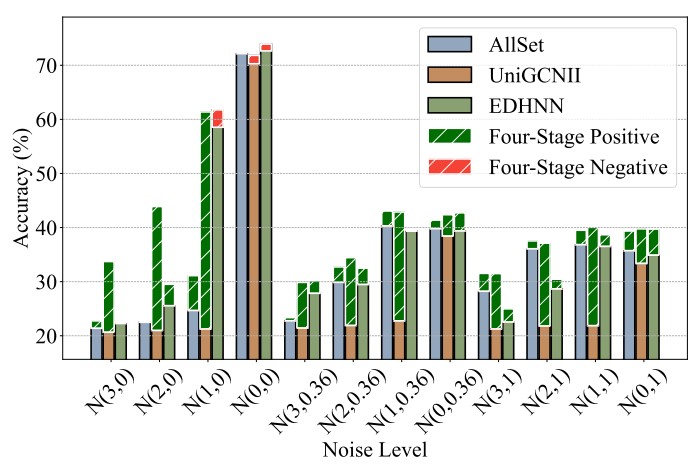

Figure 6: Performance of non-zero mean Gaussian noise in Citeseer

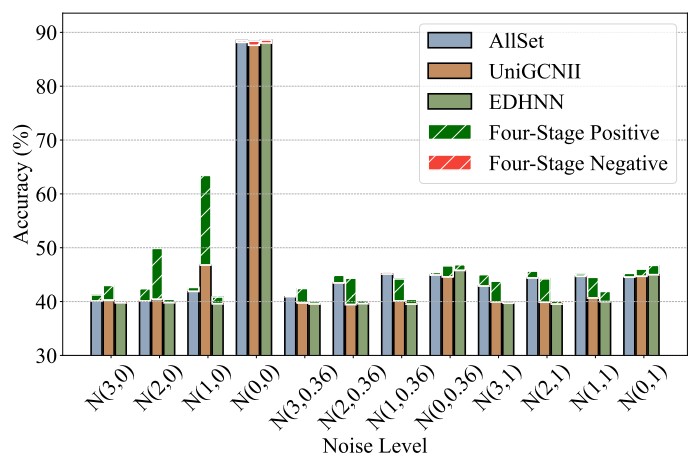

Figure 7: Performance of non-zero mean Gaussian noise in Pubmed

Among our four-stage methods, EnUniGCNII significantly outperforms others when the standard deviation $\sigma$ is 0. This observation is consistent in all tested datasets. Interestingly, when the noise mean value $\mu$ is large, e.g., 2 and 3, as the standard deviation $\sigma$ becomes larger, our models demonstrate stronger performance and denoising capability in Cora and Citeseer. For instance, consider the increasing accuracy from $\mathcal{N}(3,0)$ to $\mathcal{N}(3,0.36)$ and then to $\mathcal{N}(3,1)$ in Fig. 5. On the other hand, when fixing $\sigma$ and increasing mean value $\mu$ to larger noise, in general it results in lower accuracy, which matches with the expectation.

## A.7 Hyperedge Prediction Task

To validate the effectiveness of our proposed methods in other hypergraph tasks, we conduct hyperedge prediction experiments to evaluate the performance of three representative two-stage models and their four-stage counterparts. The hyperedge prediction task involves determining whether there exists a hyperedge among nodes in the set. We consider original hyperedges in the dataset as positive samples and generate negative samples by replacing a node in an original hyperedge with a random node not present in the hyperedge. For training, validation, and test sets we maintain positive-to-negative sample ratios of 1:5, 1:1, and 1:1 respectively. After getting node embeddings, we compute the hyperedge embedding as the sum of nodes' embedding for all nodes within a hyperedge, followed by MLP for binary classification learning. We use accuracy as the evaluation metric for hyperedge prediction. For each dataset under different noise conditions, we run 20 trials and report the mean accuracy along with the standard deviation.

Table 7: Performance in the hyperedge prediction task

| | $\mathcal{N}(0, \sigma^2)$ | ALLSETTRANSFORMER | ENALLSET | UNIGCNII | ENUNIGCNII | ED-HNN | ENEDHNN |
|---|---|---|---|---|---|---|---|
| CORA | 1 | 62.04±1.4 | 63.98±1.3 | 61.54±1.3 | 64.49±2.0 | 61.99±1.3 | **65.27±1.5** |
| | 0.6 | 62.61±1.6 | 64.51±1.3 | 60.16±1.4 | 64.81±1.6 | 62.16±1.9 | **66.87±1.4** |
| | 0 | 64.31±1.2 | 65.43±1.0 | 60.04±1.8 | **65.67±1.8** | 64.45±1.4 | **71.10±1.6** |
| CITESEER | 1 | 74.27±1.9 | 74.67±2.0 | 69.22±1.6 | 75.06±1.9 | 73.03±1.9 | **76.05±1.6** |
| | 0.6 | 74.18±2.3 | 75.04±2.1 | 68.75±1.7 | 74.68±1.8 | 74.06±1.8 | **76.66±1.9** |
| | 0 | 76.73±2.3 | 77.04±1.9 | 70.46±1.8 | 77.96±2.2 | 77.24±1.5 | **82.06±1.5** |
| CORA-CA | 1 | 68.87±1.4 | **68.91±1.9** | 61.94±1.3 | 67.61±1.9 | 65.60±1.8 | 67.23±2.2 |
| | 0.6 | 69.09±1.3 | 68.68±1.6 | 62.29±1.6 | 66.78±2.0 | 67.84±3.7 | **73.32±2.5** |
| | 0 | 69.67±1.8 | 68.96±1.9 | 59.80±1.9 | 69.01±1.9 | 77.31±2.7 | **79.86±2.9** |
| DBLP-CA | 1 | 68.47±0.5 | 67.99±0.9 | 53.70±0.4 | 70.07±1.0 | 71.05±2.6 | **72.48±4.1** |
| | 0.6 | 69.87±0.8 | 69.35±0.7 | 53.18±0.8 | 70.49±1.3 | **77.89±2.3** | 77.77±3.3 |
| | 0 | 74.12±0.7 | 73.53±0.8 | 53.10±2.6 | 75.72±0.8 | **80.77±3.8** | 79.86±2.9 |

As shown in Table 7, the experimental results verify the excellent performance of our four-stage message-passing models, particularly EnEDHNN, which achieves the highest top accuracy and has a large gap over the second-best model. EnUniGCNII and EnEDHNN achieve average performance improvements of 9% and 3% respectively over their original counterparts. The superior performance indirectly confirms the good quality of the learned node embeddings. As expected, when noise is introduced, the accuracy of almost all models deteriorates. However, we observe that the performance degradation is less severe than in node classification, ranging from a few percent to 12%. This indicates that node feature noise has a smaller impact on hyperedge prediction compared to node classification. This aligns with prior observations on graphs, which show that structural changes affect edge prediction accuracy more significantly than node classification Yang et al. (2020).

## A.8 ABLATION STUDY

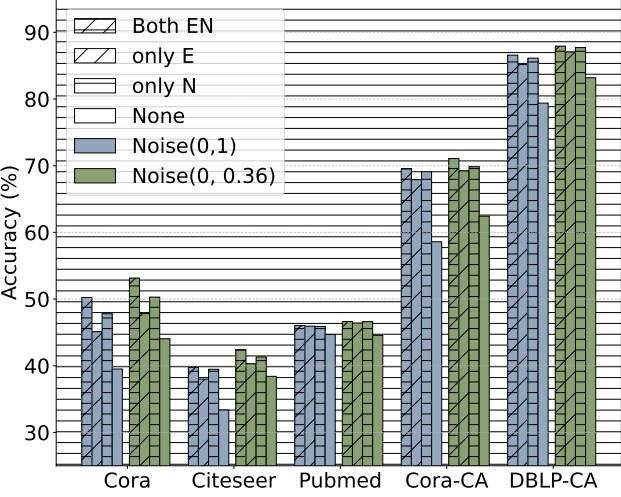

Figure 8: Results of ablation study for EnUniGCNII

To investigate the impact of different modules on enhancing denoising capability, we conduct ablation experiments on our three models EnUniGCNII, EnAllSet, and EnEDHNN, and report the results in Figures 8, 9, and 10, respectively. We refer to our model with only hyperedge communication module as only $E$, our model with only node communication module as only $N$, our full model with both as both $EN$, and the initial two-stage model as $None$.

The experimental results demonstrate that both the hyperedge and node communication modules are indispensable for enhancing accuracy. It is clear that combining both modules consistently

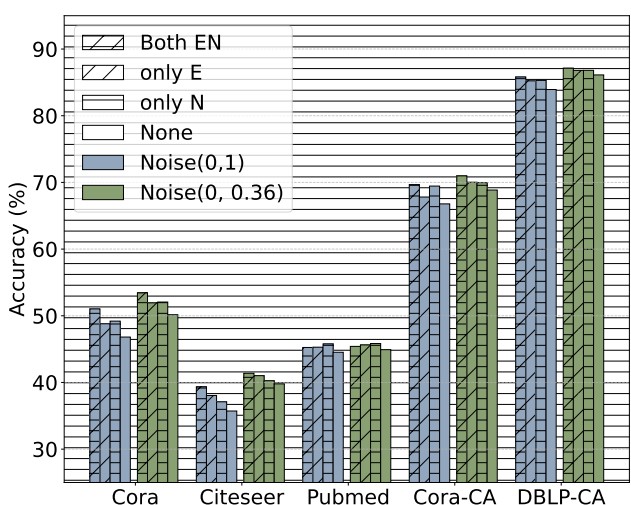

Figure 9: Results of ablation study for EnAllSet

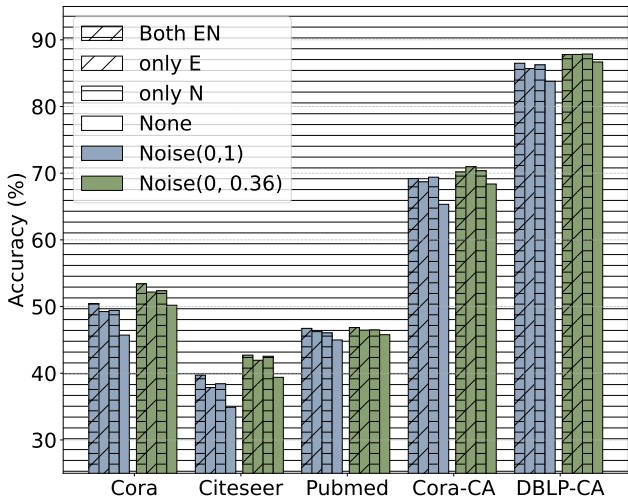

Figure 10: Results of ablation study for EnEDHNN

outperforms using any single module (only $E$ or only $N$), which further significantly improves over using none of them ($None$). Comparing the two modules, the node communication module appears to make a greater contribution in improving denoising capability because only $E$ often gets worse accuracy compared to only $N$ under both moderate and heavy noise.

### A.9 NODE EMBEDDING VISUALIZATION

In this set of experiments, we visualize the learned node embeddings (obtained from the output before the softmax function) and analyze their quality. We use t-SNE van der Maaten & Hinton (2008) to map high-dimensional representational vectors to a two-dimensional space and then plot. To match with the accuracy analysis, we choose to visualize the embeddings of nodes in the test set instead of the training set. As shown in Fig. 11, we present the node embedding visualization for EnUniGCNII, EnAllSet, EnEDHNN and their original counterparts on the DBLP-CA dataset with noise $\mathcal{N}(0, 1)$. Different colors represent different node classes. We observe that the visualizations of our enhanced embeddings in the lower row exhibit better and clearer clustering structures with fewer outliers compared to the original embeddings in the upper row (see middle parts of the figures). This experimental observation is consistent with the test accuracy for node classification shown in Table 1.

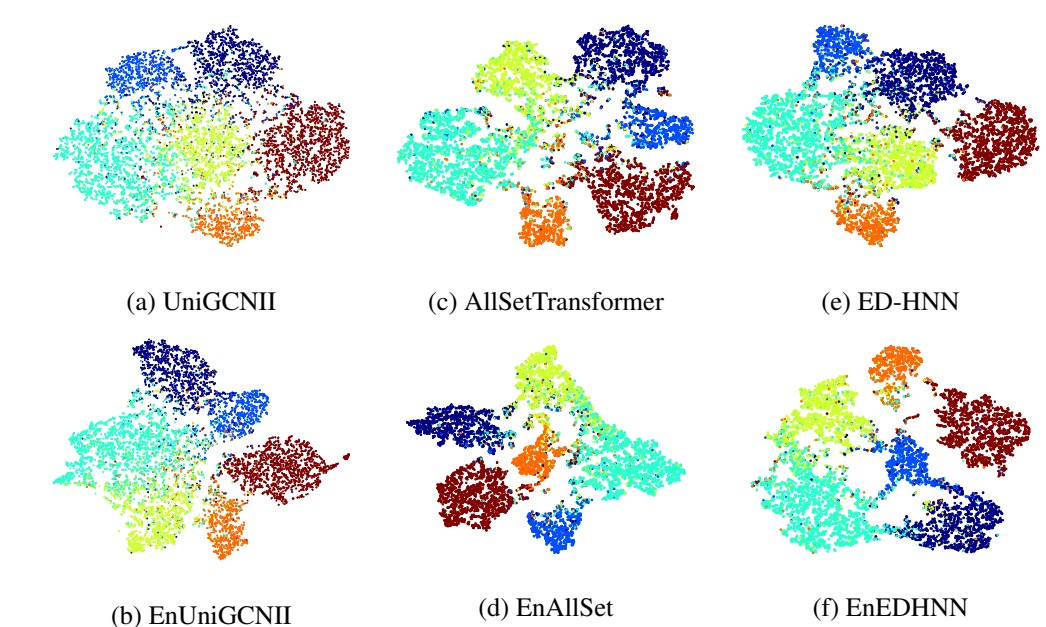

(a) UniGCNII   (c) AllSetTransformer   (e) ED-HNN

(b) EnUniGCNII   (d) EnAllSet   (f) EnEDHNN

Figure 11: Visualization of node embeddings on DBLP-CA with noise $\mathcal{N}(0, 1)$ (best seen in colors)

### A.10 HYPERPARAMETERS FOR EACH METHOD

Model hyperparameters used throughout our experiments are listed as follows. Notably, lr refers to the learning rate, h1 refers to the dimension of MLP hidden layer and heads refers to the number of attention heads. The hyperparameters of our hyperedge communication module and node communication module are aligned with the two-stage model counterparts.

Table 8: Hyperparameters of tested methods

| $\mathcal{N}(0, \sigma^2)$ | Cora | | | Citeseer | | | Pubmed | | |
|---|---|---|---|---|---|---|---|---|---|
| | lr | h1 | heads | lr | h1 | heads | lr | h1 | heads |
| MLP | 0.01 | 64 | 1 | 0.01 | 64 | 1 | 0.01 | 64 | 1 |
| HNHN | 0.001 | 512 | 1 | 0.001 | 256 | 1 | 0.001 | 512 | 1 |
| HGNN | 0.001 | 512 | 1 | 0.001 | 256 | 1 | 0.001 | 512 | 1 |
| HCHA | 0.001 | 256 | 1 | 0.001 | 128 | 1 | 0.001 | 512 | 1 |
| HyperGCN | 0.001 | 64 | 1 | 0.01 | 64 | 1 | 0.01 | 64 | 1 |
| UniGCNII | 0.001 | 512 | 8 | 0.001 | 128 | 1 | 0.001 | 128 | 1 |
| AllSetTransformer | 0.001 | 256 | 4 | 0.001 | 512 | 8 | 0.001 | 256 | 8 |
| ED-HNN | 0.001 | 128 | 1 | 0.001 | 128 | 1 | 0.001 | 128 | 1 |
| *EnUniGCNII (Ours)* | 0.001 | 512 | 8 | 0.001 | 128 | 1 | 0.001 | 128 | 1 |
| *EnAllSet (Ours)* | 0.001 | 256 | 4 | 0.001 | 512 | 8 | 0.001 | 256 | 8 |
| *EnEDHNN (Ours)* | 0.001 | 128 | 1 | 0.001 | 128 | 1 | 0.001 | 128 | 1 |
| $\mathcal{N}(0, \sigma^2)$ | Cora-CA | | | DBLP-CA | | | 20Newsgroups | | |
| | lr | h1 | heads | lr | h1 | heads | lr | h1 | heads |
| MLP | 0.01 | 64 | 1 | 0.01 | 64 | 1 | 0.01 | 64 | 1 |
| HNHN | 0.001 | 512 | 1 | 0.001 | 512 | 1 | 0.001 | 512 | 1 |
| HGNN | 0.001 | 128 | 1 | 0.001 | 256 | 1 | 0.001 | 64 | 1 |
| HCHA | 0.001 | 128 | 1 | 0.001 | 512 | 1 | 0.001 | 512 | 1 |
| HyperGCN | 0.01 | 64 | 1 | 0.01 | 64 | 1 | 0.01 | 64 | 1 |
| UniGCNII | 0.001 | 512 | 4 | 0.001 | 256 | 8 | 0.001 | 128 | 8 |
| AllSetTransformer | 0.001 | 128 | 8 | 0.001 | 512 | 8 | 0.001 | 256 | 8 |
| ED-HNN | 0.001 | 128 | 1 | 0.001 | 128 | 1 | 0.001 | 128 | 1 |
| *EnUniGCNII (Ours)* | 0.001 | 512 | 4 | 0.001 | 256 | 8 | 0.001 | 128 | 8 |
| *EnAllSet (Ours)* | 0.001 | 256 | 4 | 0.001 | 512 | 8 | 0.001 | 256 | 8 |
| *EnEDHNN (Ours)* | 0.001 | 128 | 1 | 0.001 | 128 | 1 | 0.001 | 128 | 1 |