# OpenReview forum: "On Understanding Denoising Capability in Hypergraph Representation Learning"
_ICLR.cc/2026/Conference — Submitted to ICLR 2026_

### Official Review · Reviewer_JRAx · 2025-10-30

**Soundness:** 2
**Presentation:** 3
**Contribution:** 3
**Rating:** 6
**Confidence:** 3

**Summary:**

The study investigates the effect of injecting noise into node features on the performance of hypergraph neural networks. It shows that current architectures are heavily affected by Gaussian noise and postulates that this is due to the small receptive field of the networks. The study demonstrates that deeper architectures are less affected in noisy conditions and also proposes a new framework that, in addition to the two standard stages (node-to-edge and edge-to-node), incorporates two additional stages (node-to-node and edge-to-edge). The resulting architecture shows a clear advantage in noisy settings while maintaining strong performance in the clean regime.

**Strengths:**

- As far as I am aware, there are no studies on the effect of node noise in hypergraphs, so this is a solid contribution in my opinion.

- The model pairs the analysis of denoising with a general solution that shows consistent improvement.

**Weaknesses:**

- The Hypergraph Transformer model [1] also incorporates implicit node-to-node and edge-to-edge propagation. It would be interesting to compare its performance in the denoising setup.
- While the conjecture is partially validated empirically, it is difficult to disentangle the effects of overfitting, oversmoothing, and receptive field size when applying multiple layers of a hypergraph network. Is it possible that overfitting is less pronounced in the noisy setup? For example, I wonder if sharing parameters across multiple layers affects this behavior. Additionally, is there a way to measure how noise propagates in a standard HNN versus an ENHNN to test whether the conjecture holds?

[1] Liu et al. HYPERGRAPH TRANSFORMER FOR SEMI-SUPERVISED CLASSIFICATION

Minors:

  - The format of citations makes reading difficult. Consider usin \citep instead of \cite.

  - In Equation 8 it should be an exp()

  - On row 382 there is a missing reference

**Questions:**

Please see the Weaknesses section

---

### Official Review · Reviewer_Mwkm · 2025-10-31

**Soundness:** 2
**Presentation:** 2
**Contribution:** 1
**Rating:** 2
**Confidence:** 3

**Summary:**

The present work proposes 4-stage message-passing framework for hypergraph learning. The authors find that standard 2-stage message passing frameworks for hypergraph neural networks (HNNs) miss node-to-node and hyperedge-to-hyperedge message passing. Motivated by the finding, the method design aims to improve the receptive field of the HNN layer to improve its denoising capability. In node classification and hyperedge prediction experiments, the proposed method shows superior performance to baseline HNNs.

**Strengths:**

Strengths of the present work are as follows:
- clear presentation
- development of a new method
- extensive experiments and superior performance under heavy noise
- formal analyses are provided to support some of the authors' key claims

**Weaknesses:**

I have several major concerns about the present work.

First, I find the method and analysis somewhat trivial and non-significant.
- [trivial analysis]. The connection between the receptive field and denosing capability is trivial. It is widely known that a standard graph convolution is a smoothing operator, and it serves as a low-pass filter of node signals. Thus, the proposed connection between receptive field and denosing capability is not novel. Theorem 1 and Lemma 1 seem trivial, too. Using the four-stage message-passing, obviously, should have a larger receptive field than a two-stage counterpart. I do not see how the Theorem and Lemma are surprising findings.
- [misleading assumption]. The paper argument rests on the assumption that certain node information is lost during 2-stage message passing, while a 1-stage message passing (e.g., node-to-node) may preserve it (lines 204-205). However, this assumption is unfounded. Message passing and non-linear transformation, whether it is based on a 1-stage or a 2-stage framework, aim to enrich node information, and thus, the authors' assumption that the 2-stage framework leaves only 'partial information' (line 205) is not convincing without concrete evidence.
- [trivial method]. Adding N2N (node-to-node) and H2H (hyperedge-to-hyperedge) message passing, on top of N2H and H2N message passing, is a technically trivial to improve the denoising capability of HNNs. I do not think this is a technical innovation that is typically appreciated in a top-tier conference.

The experimental outcomes also raise suspicions about the contribution of the present method.
- [marginal performance improvement]. Performance improvement is only substantial when the injected noise is very heavy. When no noise was injected, the performance of the proposed method is worse than the baseline HNNs. Moreover, after stacking multiple layers, the performance gap between the proposed method and the baselines (especially ED-HNN) is substantially reduced.

Also, I find the experimental setting problematic.
- [unrealistic noise injection]. In most of the benchmark datasets, the input node features are sparse, multi-hot vectors. However, the noise that the authors injected is Gaussian. This is highly unrealistic and, thus, problematic. The significant performance degradation of the baseline HNNs may be partially due to this unrealistic noise injection.
- [missing baselines]. If denoising capability is the major strength of the proposed method, I would expect baseline methods that improve denoising capability, including GNN-based ones (e.g., [1]). However, all the baselines are standard HNN encoders. If it turns out that GNN-based baselines have better denoising capability than the proposed method in the hypergraph benchmarks, I do not think there is a contribution of the proposed method.
- [fairness] Furthermore, fixing the number of HNN layers in the experimental comparison is unfair. Each message-passing step generally improves denoising capability, and the proposed method is designed to conduct more message passing within a single HNN layer. Then, with a fixed number of layers, it is only natural that the proposed method has better denoising capability than the baselines (since it conducts more message passing steps). For fairer comparison, the number of layers should be tuned for each HNN model, dataset, and injected noise level.

Lastly, I do not think deferring most of the main experiments, e.g., hyperedge prediction and the effect of stacking multiple layers, to the Appendix is desirable.

Overall, with the given limitations, I do not recommend accepting the paper.

Reference
- [1] Graph Neural Networks with Adaptive Residual, NeurIPS 2021

**Questions:**

See weakness

---

### Official Review · Reviewer_GRri · 2025-11-01

**Soundness:** 3
**Presentation:** 2
**Contribution:** 2
**Rating:** 4
**Confidence:** 4

**Summary:**

In this paper, the authors aim to address the challenge of noise robustness in hypergraph representation learning. By establishing a connection between receptive fields and denoising capabilities, they propose a four-stage message-passing method that expands receptive fields to mitigate the impact of noise in node features. Theoretical analysis demonstrates the increase in receptive fields, while experimental results verify the effectiveness of the proposed method under noisy conditions.

**Strengths:**

1. The paper focuses on an important topic, noise robustness in hypergraph learning.

2. The experiments show that the proposed method helps alleviate the impact of noise present in node features on hypergraph representation learning across multiple datasets.

**Weaknesses:**

1. For hypergraph representation learning, noise may arise from both hyperedges and node features, while this paper focuses only on feature noise, which is somewhat limited.

2. The technical novelty is limited, as all four message-passing steps already exist in previous message-passing neural network frameworks.

3. Although convergence times are reported, the overall computational cost and scalability should be discussed in greater depth.

4. While the paper focuses on hypergraph representation learning, experiments on larger hypergraphs are needed to further evaluate the effectiveness of the proposed method.

**Questions:**

See that in weaknesses.

---

### Official Review · Reviewer_DsAU · 2025-11-01

**Soundness:** 3
**Presentation:** 2
**Contribution:** 3
**Rating:** 4
**Confidence:** 4

**Summary:**

This paper studies the impact of noise in node features for hypergraph representation learning. The authors first connect receptive fields with denoising capability, arguing that enlarging receptive fields can improve robustness. They then propose a four-stage message-passing scheme that expands the receptive field within a single neural network layer, designed as a drop-in extension to two-stage state-of-the-art (SOTA) methods.

**Strengths:**

1. Clear core idea and generally readable presentation. The main motivation and contributions are accessible.
2. Provides theoretical analysis linking receptive field size to denoising capacity, offering a rationale for the proposed four-stage message passing.

**Weaknesses:**

1. The manuscript appears to have formatting issues (e.g., section header margins in the Experiments section). Also, there is a broken reference around line 382 pointing to a figure.
2. Missing or underdeveloped conclusion. The paper would benefit from a concise Conclusion section that summarizes findings, limitations, and future directions.
3. Scope limited to homophilic settings.The evaluated methods and datasets appear primarily homophilic. How does the proposed approach perform on heterophilic hypergraphs considered in prior work [1, 2, 3]? These datasets often include naturally noisy signals (e.g., as in AllSet’s setup [1]).
4. Fairness of comparison and parameter/complexity control. The proposed method introduces additional message-passing steps relative to two-stage SOTA baselines, which may increase depth, parameters, and computational budget. Additional experiments are needed to validate the effectiveness of the proposed method:
   - The total number parameters of the proposed method compared with SOTA baselines.
   - Ablation studies by (i) replacing the message passing mechanisms with other operations like MLPs; (ii) Compare four layers of AllSet, ED-HNN with two layers of the proposed method. These ablation studies may further empirically validate that the proposed method is not merely add more message passing layers but the designed mechanism is effective.


[1] You are AllSet: A Multiset Function Framework for Hypergraph Neural Networks. ICLR 2022.
[2] Equivariant Hypergraph Diffusion Neural Operators. ICLR 2023.
[3] Sheaf Hypergraph Networks. NeurIPS 2023.

**Questions:**

See weakness.

---

### Meta-Review · Area_Chair_qt1k · 2025-12-05

**Summary:**

Reviewers’ concerns primarily focus on misleading assumptions, trivial analysis and methods,  problematic experimental setting, unconvincing validation, and limited technical novelty. Besides, the structure and the clarity of the paper are also pointed out.    Since the authors didn’t provide any feedback,  I believe all authors will maintain their scores as (6, 4, 4, 2).  Therefore,  I think this paper can be rejected.

**Reviewer Concerns:**

None of the concerns were addressed, since the authors didn’t provide any feedback.

**Reviewer Scores:**

Since the authors don’t provide any feedback, I believe all authors will maintain their scores as follows
- JRAx		Rating: 6 / Confidence: 3
- GRri		Rating: 4 / Confidence: 4
- Mwkm		Rating: 2 / Confidence: 3
- DsAU		Rating: 4 / Confidence: 4

---

### Decision · Program_Chairs · 2026-01-26

Reject